# Towards Automatic Discovery and Explanation of Differences Between Vision Models

## Abstract

Researchers and developers often compare state-of-the-art and newly developed models beyond benchmark scores, using techniques such as visualizations, case-by-case analyses, and qualitative evaluations. Such analyses provide deeper insights into model behaviors and often motivate the development of improved models and the establishment of new benchmarks. However, identifying strengths and weaknesses typically requires extensive human effort, consuming a significant amount of time and resources. To address this challenge, we explore the automatic generation of natural language explanations that describe the performance differences between two models. We introduce three evaluation metrics for explanations: `Completeness` for correctness and overall informativeness, `Density` for token-level informativeness, and `Token Length` for the verbosity of explanations. Building on these metrics, we propose three explanation generation methods: `Raw Differences`, which enumerates all performance differences; `Summarization`, which condenses them into concise summaries; and `Optimization`, which optimizes explanations for both informativeness and conciseness. We evaluate our framework on CMNIST, CLEVR, and CelebA, showing that `Optimization` effectively uncovers model differences and biases in natural language. For reproducibility, we will release the code and data.

## 1 Introduction

Despite the prevalence of standardized benchmarks for model evaluation (Deng et al., 2009; Lin et al., 2014), researchers often conduct additional analyses such as visualizations or case studies (Naseer et al., 2021). These reveal strengths and weaknesses overlooked by aggregate metrics, guiding benchmark design (Liu et al., 2024b) and inspiring new methods (Sagawa et al., 2020). However, such analyses are typically ad hoc, labor-intensive, and difficult to scale.

Our work aims to reduce or replace this process using foundation models and synthetic data. Recent advances show that large language models (LLMs) (Grattafiori et al., 2024; Hurst et al., 2024) can substitute for human evaluators (Chiang & Lee, 2023; Bills et al., 2023) and, when used as agents, even perform autonomous decision-making (e.g., LangChain). Synthetic data, meanwhile, provides controllable resources for training models to address weaknesses (Kim et al., 2024a) and enabling more detailed evaluations (Geirhos et al., 2018). Figure 1 shows our framework for automatically comparing two vision models and explaining their differences in natural language: instead of a human, an LLM probes models with a generator and produces concise explanations.

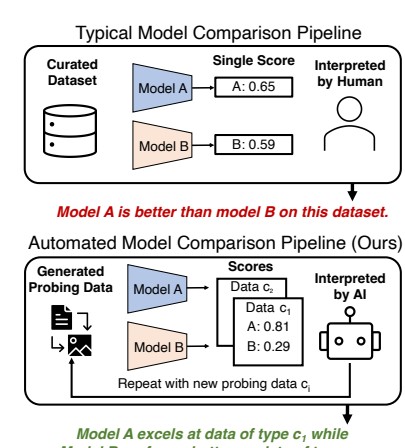

Figure 1: Given vision models, A and B, our method for explanations leverages an LLM to probe their predictions iteratively with the help of a generator.

We first introduce three metrics for evaluating explanation quality. `Completeness` measures whether an explanation provides sufficient information, with higher values indicating accurate

reasoning about model predictions from the explanation alone. `Density` quantifies how much `Completeness` drops when tokens are randomly removed, indicating how informative each token is. We also report `Token Length` to measure verbosity, since lengthy explanations are undesirable. Together, these metrics provide a comprehensive assessment of explanation quality.

We then propose one baseline and two methods for generating comparative explanations. First, the `Raw Differences` baseline directly lists performance differences of two models across all conditions. While comprehensive, it becomes overwhelming as conditions grow and fails to highlight critical insights. To address this, `Summarization` uses an LLM-based summarization module to condense listings into a concise and insightful explanation. However, summaries may omit subtle details and rely on the LLM's summarization capability. To overcome these issues, we propose `Optimization`, which optimizes explanations to be both correct and concise. Following prior work on text optimization (Yuksekgonul et al., 2025; Xiao et al., 2025; Khattab et al., 2024), we iteratively refine explanations using LLM feedback, preserving the coverage of `Raw Differences` while gaining the conciseness of `Summarization`.

We evaluate our methods on CMNIST (Arjovsky et al., 2019), CLEVR (Johnson et al., 2017), a synthetic gender dataset [1], and CelebA (Liu et al., 2015).Experiments show that our methods reveal true differences between vision models, and performance gains from explanations further validate their effectiveness. Our contributions are: (1) three metrics for evaluating explanations of model differences; (2) an automatic framework for generating natural-language explanations of model differences; (3) extensive experiments demonstrating the effectiveness of our methods.

## 2 RELATED WORK

**Comparative Analysis.** Many benchmarks (Deng et al., 2009; Lin et al., 2014) evaluate model performance and represent it with a single compressed score, enabling direct comparison across models by ranking. Such comparisons help assess whether new models improve upon previous ones, provide insights for refinement, and guide model selection for deployment. However, a single score cannot capture the multifaceted nature of models (Geirhos et al., 2018; 2020). For example, improvements in fairness are often overlooked. As a result, researchers turn to qualitative analyses to study differences, which motivates the development of new benchmarks and models that address diverse perspectives (Sagawa et al., 2020). Yet such analyses are labor-intensive and time-consuming, as they require human effort. To overcome these limitations, we propose an automated framework for explaining prediction differences between vision models.

Several comparative analysis methods have been proposed. Jhamtani & Berg-Kirkpatrick (2018) describe differences between image pairs, while Dunlap et al. (2024) focus on image sets. Chiquier et al. (2025) generate images with subtle differences while preserving identity. VibeCheck (Dunlap et al., 2025) evaluates vibe differences between LLM outputs. In this work, we aim to compare two vision models and generate concise explanations, along with metrics to evaluate the quality of these explanations. Our method and metrics build on synthetic data generation and LLMs.

**Synthetic Data.** Synthetic data has long been used for evaluation (Hendrycks & Dietterich, 2019; Mayer et al., 2016) and training (Tobin et al., 2017; Johnson et al., 2017), valued for its scalability and manipulability. With advances in generative models such as diffusion models, synthetic images now achieve unprecedented quality, spurring new applications (Kim et al., 2024a; Ye-Bin et al., 2024; Augustin et al., 2022; Jeanneret et al., 2022). We leverage Blender (Blender Online Community, 2025) and diffusion models (Esser et al., 2024) to analyze models without relying on predefined image datasets.

**Textual Optimization.** Retraining LLMs is computationally expensive; therefore, many approaches instead optimize the input text. Methods such as TextGrad (Yuksekgonul et al., 2025), DSPy (Khattab et al., 2024), and Verbalized Machine Learning (Xiao et al., 2025) adapt text prompts to achieve task-specific goals. Building on this work, we optimize explanations that capture prediction differences between two vision models. Our framework enables the LLM to iteratively refine explanations, incorporate feedback, and determine which conditions to probe two vision models, thereby improving the quality of the explanation.

---

[1]This dataset is constructed for the explanation evaluation on the proposed metrics.

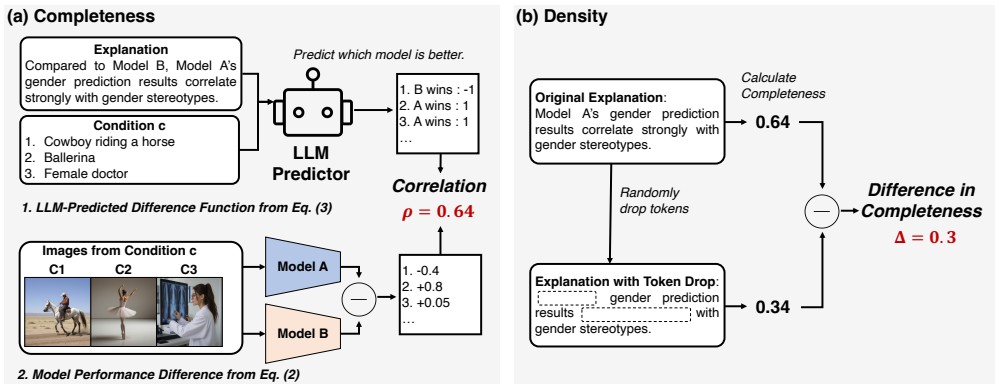

Figure 2: `Completeness` and `Density` Metrics. `Completeness` measures the correlation between the true difference of the two models' predictions and LLM predicted differences based on the explanation for the same set of data conditions. A higher value indicates that the explanation enables the LLM to reliably recover the true model differences. `Density` measures the change of `Completeness` after removing word tokens of the explanation randomly. A higher value indicates that the explanation has high information density.

**LLM Evaluator.** LLMs are increasingly employed as evaluators to reduce time, labor, and cost (Hackl et al., 2023; He et al., 2024; Liu et al., 2023). He et al. (2024) use LLMs as annotators, while Liu et al. (2023) propose an LLM-based evaluation framework for natural language generation. Bills et al. (2023) simulate neural activations using LLMs. Similarly, we use LLMs to evaluate explanations. If the explanations are sufficiently complete and concise, an LLM can correctly answer explanation-related questions.

## 3 EVALUATING EXPLANATIONS OF PREDICTION DIFFERENCES BETWEEN TWO MODELS

**Setup of Comparison of Two Vision Models.** We are given two models, $\{f_A, f_B\} : \mathcal{X} \to \mathcal{Y}$, and a conditional generator, $\mathcal{G} : \mathcal{C} \to \mathcal{X} \times \mathcal{Y}$, where $\mathcal{X}$ is the input of the models, $\mathcal{Y}$ is the corresponding label, and $\mathcal{C}$ is the condition. For image tasks, the generator can be a conditional data generator, such as a Blender (Blender Online Community, 2025) or a text-to-image (T2I) diffusion model. Suppose that `Explainer` is an algorithm that creates an explanation describing how two models' predictions differ in the form of natural language. Then, we formulate the process as follows:

$$\text{Explanation} = \text{Explainer}(f_A, f_B, \mathcal{G}), \tag{1}$$

where `Explainer` has access to both of the models and a data generator to produce an explanation.

**Explanation Completeness Score.** A good explanation is one that fully expresses the phenomenon in natural language and allows one to answer new questions about the phenomenon correctly. Moreover, a good explanation should be able to accurately predict which models will perform better on a new, unseen sample, even before running any inference with the models.

To measure whether the explanation accurately approximates the models' behaviors, we use an LLM that is fed the explanation as a proxy model and measure `Completeness` by comparing the proxy model's outputs with those of the actual models for probing data. In the following, we first define two functions representing the actual models' performance difference and LLM prediction, and then formally define `Completeness`. We represent the model performance difference as follows.

**Definition 1.1. (Model Performance Difference Function)** Let $f_A$ and $f_B$ be two models to be compared. Given a condition $c$ and corresponding data $\{x_i^c, y_i^c\}_{i=1}^{i=n} \sim \mathcal{G}(c)$, we define the model performance difference function as:

$$\text{Diff}_{\text{Model}}(f_A, f_B, c) = \text{Perf}(f_A, \{x_i^c, y_i^c\}_{i=1}^{i=n}) - \text{Perf}(f_B, \{x_i^c, y_i^c\}_{i=1}^{i=n}), \tag{2}$$

where `Perf` denotes the performance, *e.g.*, accuracy, on the given data with condition $c$.

The condition $c$ can be any characteristic of the data that produces a subset of the data distribution with the generator $\mathcal{G}$. The model performance difference function $\texttt{Diff}_{\texttt{Model}}(\cdot)$ is positive if model $f_A$ achieves a higher accuracy, negative if model $f_B$ performs better, and 0 otherwise. We leverage the reasoning capabilities of an LLM together with the explanation to create a proxy model that predicts these model performance differences. Given an explanation and a condition $c$, we prompt the LLM to predict which model would perform better.

**Definition 1.2. (LLM-predicted Difference Function)** Let $c$ be the condition defining a subset of the data, and $o$ be the output of the LLM prompted to decide on the better model based on the explanation and $c$. We define the LLM-predicted difference function as:

$$\texttt{Diff}_{\texttt{LLM}}(c; \texttt{Explanation}) = \begin{cases} 1 & \text{if} \quad o = \text{``Model A is better''}, \\ 0 & \text{if} \quad o = \text{``Cannot be determined''}, \\ -1 & \text{if} \quad o = \text{``Model B is better''}. \end{cases} \quad (3)$$

We define the `Completeness` metric using the correlation between the LLM's answers and the actual model differences.

**Definition 1. (Completeness)** Given $\texttt{Diff}_{\texttt{LLM}}$ and $\texttt{Diff}_{\texttt{Model}}$, we define `Completeness` of an explanation as the correlation between the two functions:

$$\texttt{Completeness} = \texttt{correlation}_{\mathcal{C}}(\texttt{Diff}_{\texttt{Model}}, \texttt{Diff}_{\texttt{LLM}}). \quad (4)$$

A higher correlation indicates a better explanation because it enables an LLM to predict the correct outcome based solely on the explanation more frequently. A correlation of 1 across all samples means that the model difference can be perfectly predicted based solely on the explanation. The conditions $c$ on which $\texttt{Diff}_{\texttt{Model}}$ is evaluated come from a pre-defined test set of textual conditions. If such a set of conditions is not available, vision models can be evaluated by captioning each test image and using the caption as $c$. Figure 2 summarizes the steps of computing `Completeness`.

**Density Score.** This metric is defined by computing counterfactual changes of `Completeness` after perturbation: "What if a subset of the explanation is removed? Could an LLM still answer correctly?" Based on this criterion, the tokens of an explanation can be categorized based on the change of `Completeness` score after token removal ($\Delta$): unnecessary (removal does not change the score, $\Delta = 0$), informative (removal decreases the score, $\Delta > 0$), and misleading (Removal increases the score, $\Delta < 0$). A higher `Density` indicates that many tokens are informative, reflecting greater information density. Perturbations are introduced by randomly removing tokens from the explanation for each question, and the resulting changes are aggregated across questions.

**Definition 2. (Density)** We define `Density` of an explanation as:

$$\texttt{Density} = \texttt{Completeness} - \widehat{\texttt{Completeness}}, \quad (5)$$

where $\widehat{\texttt{Completeness}}$ is computed from the explanation with randomly dropped tokens. Specifically, Completeness is evaluated for each condition c defined in Eq. (3), and the Density captures how much the Completeness degrades under such perturbations.

**Token Length.** Lengthy and verbose explanations are harder to interpret and are more likely to include redundant words. Therefore, we also report the number of tokens as an indicator of conciseness. Together, Completeness, Density, and the Token Length capture complementary aspects of an explanation. While each metric focuses on a different dimension, considering them jointly provides a more comprehensive understanding of explanation quality.

## 4 AUTOMATIC DISCOVERY OF MODEL DIFFERENCES

Now that we have established two metrics for evaluating textual explanation, we propose three methods to create the explanations as in Eq. (1), with access to the two models, $f_A$ and $f_B$, along with the conditional generator $\mathcal{G}$. We employ LLM to generate textual explanations.

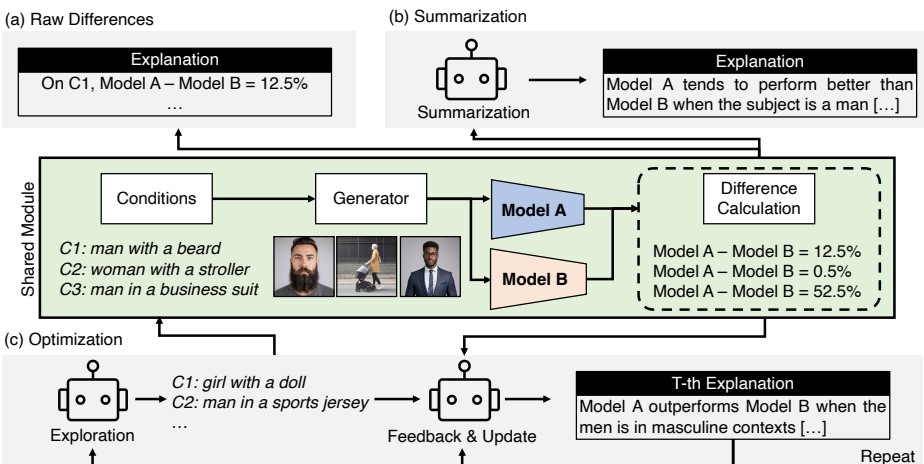

Figure 3: **Three Methods for Explanations.** **(a)** `Raw Differences` aggregates performance differences across conditions without losing information. **(b)** `Summarization` condenses the results from all conditions into a single paragraph to reduce token length. **(c)** `Optimization` iteratively refines the explanation, ensuring that no information is lost.

**Raw Differences.** A concept-based approach inspires the first baseline. If the concepts within images are known, we can evaluate the model's performance on each concept to understand its behavior better. Figure 3 (a) illustrates the pipeline of `Raw Differences`.

First, we sample conditions that define subsets of the data distribution, e.g., attributes of a person that a gender classifier might be biased towards. These can either be drawn randomly from known set of conditions or generated by an LLM to obtain open-set conditions. Next, we use a conditional generator to produce images based on the sampled conditions and measure the performance differences between models, $f_A$ and $f_B$. Finally, the explanation is given as a list of these performance differences, one for each condition. The main advantage of this approach is that it avoids information loss. If both humans and LLMs can correctly interpret the large amount of comparison data, the explanation can clearly convey differences in model behavior. However, the drawback is that the explanations become lengthy.

**Summarization.** To overcome the limitations of the above method, we introduce a summarization module. Recent advances in LLMs have shown strong performance across diverse language tasks, including condensing long documents into concise summaries (Zhang et al., 2024; Liu et al., 2024a). We leverage this capability by applying LLM-based summarization to the output of `Raw Differences` as shown in Fig. 3 (b). The key advantage of `Summarization` is its shorter explanation length compared to `Raw Differences`, while still preserving essential information when the summarization is effective. However, a drawback is that critical cues may be lost if LLM summarization is not perfect.

**Optimization.** We introduce an explanation refinement method to ensure that discovered explanations are both complete and concise. As LLMs can handle diverse tasks when guided by appropriate prompts, considerable work has focused on optimizing how language is given to LLMs. Approaches such as TextGrad (Yuksekgonul et al., 2025), DSPy (Khattab et al., 2024), and Verbalized Machine Learning (Xiao et al., 2025) demonstrate the effectiveness of prompt optimization. We adapt this idea to our task as follows:

1. *Exploration*: The LLM proposes new conditions to explore in order to improve the explanation.

2. *Feedback/Update*: Based on the outcomes under these conditions, the LLM provides feedback on how to refine and update the explanation. Repeat steps 1-2 as in Fig. 3.

Step 1 (Explore conditions) can be viewed as generating probing samples: the LLM proposes candidate conditions, analogous to drawing data points from an open set of evaluation data for better understanding the models. Step 2 (Update explanation with feedback) then functions as a reasoning step to describe the model differences more effectively: the LLM evaluates these conditions, produces feedback, and refines the explanation accordingly. We define the objective function as the

Table 1: **Scores on CMNIST.** We construct the human-written explanations under the assumption that we already know how to train models, and denote these as Humans, which serve as the upper bound. For automatic explanations, we adopt Llama 3.1 8B (Grattafiori et al., 2024) and Phi 4 14B (Abdin et al., 2024). We adopt GPT-5 mini as an evaluator. We find that iterative refinement, `Optimization`, consistently outperforms other automatic explanation methods.

| LLM | Method | Completeness | Density | Token Length |
|---|---|---|---|---|
| - | Human | 0.90 | 0.51 | 74 |
| - | Raw Differences | 0.33 | 0.15 | 2813 |
| Llama 3.1 8B | Summarization | 0.55 | 0.23 | 130 |
| | Optimization | 0.66 | 0.28 | 61 |
| Phi 4 14B | Summarization | 0.58 | 0.03 | 105 |
| | Optimization | 0.67 | 0.23 | 71 |

sum of `Completeness` and `Density`. At each iteration, we start with $n$ candidate explanations. Each of them is refined through steps 1-2, producing $n$ updated explanations. Among these $2n$ explanations, we retain $n$ explanations with the highest objective scores, where the objective function is evaluated over the explored conditions. We set $n$ to 3.

We design prompts for the Exploration and Feedback/Update steps. Each prompt assigns the role of a machine learning researcher to the LLMs, a widely adopted strategy for guiding the behavior of LLMs. To refine explanations, the prompts explicitly emphasize conciseness. For example, we prevent the model from simply enumerating accuracy differences. Prompts also incorporate the proposed metrics and task descriptions to ensure that the LLMs understand the intended objectives and can reason about the best next condition to explore or how to update the explanation effectively.

In summary, `Raw Differences` provides a high level of information but tends to produce lengthy explanations. `Summarization` yields concise explanations, but the important information could be removed. In contrast, `Optimization` maintains a high level of information while also generating concise explanations, thereby combining the strengths of the other two approaches. Please refer to Appendix A for additional details, including LLM prompts and pseudocode for all methods.

## 5 EXPERIMENTS

### 5.1 CMNIST EXPERIMENT

**Setup.** We construct two biased models on CMNIST (Arjovsky et al., 2019; Bahng et al., 2020), a colored variant of MNIST, to compare image classifiers. In addition to digit and color, we introduce a distracting factor, rotation, that is irrelevant to model performance. Specifically, Model A ($f_A$) is trained on digits 0–4 with red color and digits 5–9 with all colors, while Model B ($f_B$) is trained on digits 0–4 with all colors and digits 5–9 with blue color. To estimate the upper bound of the task, we also provide explanations written with full knowledge of these biases (Human) shown in Fig. 4. The number of conditions observed for the three methods is 100. We randomly sample 100 conditions from all possible attribute combinations to obtain the raw differences for `Raw Differences` and `Summarization`. The number of iterations for `Optimization` is 10; the LLM can freely choose 10 different attribute combinations for each iteration during exploration.

**Quantitative Results.** Table 1 shows the performance on our proposed metrics. The LLM column specifies which model was used to generate the explanations, while evaluation is consistently conducted with GPT-5 mini. A human-written explanation achieves the highest scores (Completeness: 0.90, Density: 0.51) with 74 tokens, as experts with knowledge of the models write it. In contrast, `Raw Differences` yields low scores (Completeness 0.33, Density 0.15) despite covering many performance cases, primarily due to its excessive length (2813 tokens). `Summarization` improves both Completeness (0.55/0.58) and length (105/130 tokens), demonstrating that condensing explanations enhances their effectiveness. Finally, `Optimization` achieves the best results, reaching Completeness 0.66/0.67 and higher Density (0.23/0.28) with concise explanations (61/71 tokens). These results confirm that optimizing explanations leads to more faithful and compact representations of model behavior.

**Human**

Model A is worse than model B when the digits of 0, 1, 2, 3, and 4 are not colored in red regardless of the rotation angle. Model B is worse than model A when the digits of 5, 6, 7, 8, and 9 are not colored in blue regardless of the rotation angle.

**Summary**

Model A performs well when the digit is 9, color is magenta or grey, and angle is within a certain range. However, Model A struggles with digits 0, 4, and 2, especially when the color is yellow, green, or cyan, and the angle is outside of a specific range. Model B performs better with digits 0, 4, and 2 in various color and angle combinations, but its accuracy drops when the digit is 9, color is magenta or grey, and angle is within a specific range. Both models have strengths and weaknesses, and their performance varies depending on the input conditions.

**Optimization**

Model A and Model B show varied performance under different conditions. While A underperforms with digit 1 and certain angles, it outperforms with digits 5, 6, 7, and 8. Model B shows relative stability with certain digits and conditions but underperforms with others.

Figure 4: **Attribution Score.** We compute token attribution scores by evaluating the loss change under a leave-one-out strategy. For fair comparison, the scores are normalized to lie between 0 and 1 using the same normalization factor across tokens. We observe that the informative tokens, *e.g.*, digits, color, and performance, are highlighted.

Table 2: **Ablation Study.** Concise Prompt indicates whether the prompt guides to generate concise explanations. Metrics for Optimization specifies the objective used during optimization. For example, when both metrics are employed, optimization is performed to minimize their combined value.

| Metrics for Optimization | | Concise Prompt | Completeness | Density | Token Length |
|---|---|---|---|---|---|
| Completeness | Density | | | | |
| ✓ | ✗ | ✗ | 0.70 | 0.11 | 304 |
| ✓ | ✓ | ✗ | 0.64 | 0.18 | 209 |
| ✓ | ✗ | ✓ | 0.63 | 0.15 | 81 |
| ✓ | ✓ | ✓ | 0.66 | 0.28 | 61 |

**Attribute Score.** We compute token attribution, which is the importance of each token on explaining the model differences. Specifically, we measure the change in loss[2].under the Leave-One-Out approach. The influence scores are normalized to the range [0, 1] with the same normalization value. As shown in Fig. 4, tokens with high scores correspond to key cues, such as digits and performance-related expressions of the explanations. To further analyze attribution, we introduce the effective token ratio, which measures the proportion of tokens whose normalized influence score exceeds a given threshold. We vary this influence-score threshold in increments of 0.05 and compute the corresponding token ratios to capture how densely informative tokens are distributed within the explanation. Figure 5 shows the results. We observe

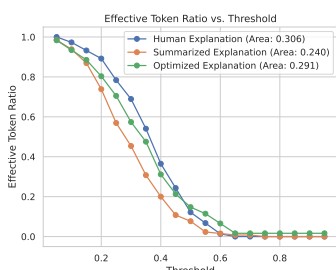

Figure 5: **Effective Token Ratio.** The curves show the proportion of tokens exceeding a given threshold from Fig. 4. The area under each curve (AUC) is reported in the legend.

that the curve of a good explanation lies higher. The area under the curve (AUC) for Human, Summarization, and Optimization is 0.306, 0.240, and 0.291, respectively. This trend is consistent with Density, since both metrics measure the information density within the explanation.

**Ablation Study.** To validate the design choice of Optimization, we conduct an ablation study, as shown in Table 2. The objective function is defined as the sum of Completeness and Density. In the metrics columns, ✓ and ✗ indicate whether the corresponding metric is included in the sum. The Concise Prompt column indicates whether an instruction to write updates concisely and clearly was included when the LLM revised the explanation. Prompt details are included in Appendix A.

The results highlight the tradeoffs between completeness, density, and token length. When optimizing only for completeness without a concise prompt, the explanation achieves the highest complete-

---

[2]We use a loss function that resembles the Completeness metric as explained in Appendix A.

Table 3: **Scores on Clevr.** To discover an explanation, we adopt Llama 3.1 8B and Phi 4 14B. We leverage GPT-5o mini as an evaluator. The tendency is same to Table 1

| LLM | Method | Completeness | Density | Token Length |
|---|---|---|---|---|
| - | Raw Differences | 0.31 | 0.13 | 600 |
| Llama 3.1 8B | Summarization | 0.22 | 0.05 | 127 |
| | Optimization | 0.40 | 0.09 | 60 |
| Phi 4 14B | Summarization | 0.10 | -0.03 | 69 |
| | Optimization | 0.30 | 0.07 | 133 |

| [...] Model B performs well with cylinders, especially when they are medium-sized and made of metal, but falters with spheres, especially when they are small and made of rubber. [...] | **More Iterations** → | [...] Model B performs well with typical metal shapes and standard sizes, but struggles with rubber shapes and non-standard sizes. [...] |
|---|---|---|
| **Score: 0.40 /. 0.09 / 60** | | **Score: 0.52 / 0.33 / 66** |

Figure 6: **More Iterations Result on Clevr.** The scores are Completeness / Density / Token Length. We observe consistent improvements as the number of iterations increases. Moreover, the written explanations become more accurate.

ness (0.70) but suffers from low density (0.11) and excessive length (304 tokens). Adding density to the objective improves density (0.18) and shortens the explanation (209 tokens), but reduces completeness (0.64). Introducing a conciseness prompt alone reduces the output (from 81 tokens) while moderately improving density (from 0.15). Finally, combining both density and concise prompt yields the best balance: completeness remains competitive (0.66), while density reaches the highest value (0.28) with the shortest length (61 tokens). These results demonstrate that Density and the concise prompt complement each other, producing compact yet informative explanations.

## 5.2 CLEVR EXPERIMENT

**Setup.** We construct two biased models on CLEVR (Johnson et al., 2017), trained to predict the shapes of geometric objects in the scene. Each image in CLEVER contains one object that is characterized by four attributes: shape (cube, cylinder, sphere), material (metal, rubber), color (gray, red, blue, green, brown, purple, cyan, yellow), and size (small, medium, large). $f_A / f_B$ are biased toward rubber/metal materials, i.e., they perform well on one material but poorly on the other. The number of conditions observed for the three methods is 20. The number of iterations for Optimization is 2 with 10 conditions per iteration.

**Results.** We observe that Raw Differences achieves higher Completeness and Density scores compared to Summarization. This is because Summarization primarily focuses on describing size, which is not a critical factor for distinguishing differences between $f_A$ and $f_B$. In the Phi-4 row, Summarization even produces a negative Density value, indicating that the explanation includes incorrect or irrelevant information. As shown in Fig. D of the appendix, its explanation mistakenly states that $f_A$ handles rubber poorly while $f_B$ excels at it, which contradicts the actual material bias. Optimization consistently yields high Completeness and low Token Length values.

To examine whether additional optimization yields further benefits, we extend Optimization from 2 to 10 iterations as shown in Fig. 6. The completeness and density scores improve from 0.40/0.09 to 0.52/0.33, while the explanation length remains nearly unchanged. The resulting explanations more clearly articulate the material bias. This indicates that the Optimization continuously improves by scaling the number of iterations as more conditions can be explored to discover better explanations.

## 5.3 EXAPLANATION OF ZERO-SHOT CLASSIFICATION OF VISION-LANGUAGE MODELS

**Setup.** To test a more realistic setting, we perform experiments on gender classification using a zero-shot classifier, i.e., SigLIP (Zhai et al., 2023). To create differences between the two classifiers, we

**Model A** outperforms Model B when the image condition involves men in **professional or stereotypically masculine contexts**, such as a man in a suit, man holding a briefcase, or man with a beard. In contrast, **Model B** performs **better in scenarios involving women or children**, such as a girl playing with dolls, woman with a purse, or woman with a baby. Model A and Model B **perform equally well in conditions with neutral or ambiguous gender cues**, like a boy riding a bike, young girl smiling, or boy playing soccer.

*Ask GPT-5 to improve prompt based on explanation*

**New prompts generated from GPT-5 given explanation**

For Model A: Since it favors masculine/professional cues, [...]
- a man casually dressed in everyday clothing, smiling in a park
- a woman dressed in professional attire, confidently walking in an office

For Model B: Since it favors women/children, [...]
- a man playing with a child at home, wearing casual clothes
- a woman wearing neutral clothing, speaking at a conference with a laptop nearby

|         |       | Avg         | Worst        | Gap          |
|---------|-------|-------------|--------------|--------------|
| Model A | Orig. | 86.7        | 58.3         | 28.4         |
|         | GPT-5 | 87.2 (0.5)  | 75.4 (17.1)  | 11.8 (16.6)  |
| Model B | Orig. | 81.5        | 14.6         | 66.9         |
|         | GPT-5 | 87.1 (5.6)  | 58.3 (43.7)  | 28.8 (38.1)  |

Figure 7: **Performance Improvement From Explanation.** We provide the discovered explanation to GPT-5 and ask it to suggest ways to improve the vision models. Based on the explanation, GPT-5 generates new prompts. We observe a mitigation of bias, which further validates the effectiveness of the discovered

use different prompts for classifying man and woman. For $f_A$, we use "a photo of a man with black hair" and "a photo of a woman with blond hair"; for $f_B$, we adopt "a photo of a man with blond hair" and "a photo of a woman with black hair". All methods employ Stable Diffusion 3.5 (Esser et al., 2024) as a data generator. The LLM explores open-set conditions, which substantially increase the difficulty of the task. For evaluation, we construct a synthetic dataset of men and women based on 50 captions per gender to measure our proposed metrics. The number of conditions observed for the three methods is 100. The number of iterations for `Optimization` is 10. Further details are provided in Appendix A.

**Results.** Table 4 shows scores on our synthetic gender dataset. `Optimization` provides the best trade-off between all three metrics. Unlike CMNIST, gender prediction requires more sophisticated reasoning because the search conditions are unconstrained and the correlations are subtle. Our primary objective is to gain a deeper understanding of the differences between models, which can guide improvements to these models.

Table 4: **Scores on Gender.** We adopt Llama 3.1 8B for generating explanations.

| Method          | Completeness | Density | Token Length |
|-----------------|--------------|---------|--------------|
| Raw Differences | 0.11         | 0.05    | 2375         |
| Summarization   | 0.11         | -0.05   | 100          |
| Optimization    | 0.17         | 0.01    | 107          |

To examine whether the discovered explanations can also guide model improvement in practice, we further conduct experiments on CelebA, a widely used dataset for gender prediction that is known to exhibit strong correlations with hair color. On CelebA, we measure the average accuracy, the worst-case accuracy across hair-color subgroups, and the performance gap between groups.

We provide the explanation from `Optimization` to GPT-5 and ask it to create new prompts that address the weaknesses of both models in order to enhance gender classification performance. Figure 7 shows the explanation discovered by `Optimization`, GPT-5's response, and the resulting performance changes on CelebA. The new prompts from GPT-5 mitigate the discovered weaknesses of the models. When using the prompts, we observe consistent improvements in average accuracy, worst-case subgroup performance, and inter-group gap. The positive results further validate the quality of the discovered explanation.

## 6  CONCLUSION

To the best of our knowledge, we present the first study on explaining vision model differences in natural language. We introduce evaluation metrics that capture the desirable properties of an explanation: informativeness and conciseness. We propose methods for generating textual explanations of model differences. Among them, `Optimization` achieves the best performance by integrating the advantages of the other approaches. We further demonstrate that the discovered explanation can mitigate the weakness of vision models, which validates the effectiveness of the explanation. Our metrics and methods are validated on vision classification tasks, and extending them to other domains and modalities represents a promising direction for future research.

## REPRODUCIBILITY STATEMENT

For reproducibility, we provide detailed setups in Appendix A. Specifically, Appendix A.1 describes the prompts and hyperparameters used in the evaluation metrics. In Appendix A.2, prompts and pseudocode for each method are given. Appendix A.3 explains the datasets and provides sample instances, and Appendix A.4 reports the computing resources and hyperparameters used with LLMs. We plan to release the code and data publicly upon acceptance of the paper for reproducibility.

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

# Appendix

## USE OF LARGE LANGUAGE MODELS (LLMS)

We made use of large language models (LLMs) to assist in refining the writing of the paper. In addition, LLMs provided support in improving the clarity of the writing and offering guidance on LaTeX usage and formatting.

## A    DETAIL AND SETUP

### A.1    EVALUATION

**Completeness.** We evaluate `Completeness` by measuring the correlation between ground-truth answers (Eq. (2)) and LLM predictions (Eq. (3)). A high-quality explanation should enable the LLM to give the correct answer. To this end, we adopt LLMs as evaluators, leveraging their strong reasoning capabilities. Such use of LLMs, often referred to as LLM judges, has become common in prior work (Verga et al., 2024; Kim et al., 2024b; Hackl et al., 2023; He et al., 2024; Liu et al., 2023); the key advantage is automatic and scalable evaluation. We use the following prompt template to get the LLM prediction given an explanation:

> You are a machine learning researcher. Model A and Model B are {Task Description}. You will be given an explanation that describes model A and model B. Given the explanation and corresponding question, you need to choose an answer from the options.
>
> [Example]
> {In-Context Example}
>
> Now, let's start the evaluation.
> Explanation: {Explanation}
> Question: {Question}
> Options: [1] Model A, [2] Model B, [3] Cannot be determined
> Answer:

{Task Description} specifies the models' task (e.g., classifying digit images from 0 to 9). {In-Context Example} provide the task information to LLM explicitly. We give one example to LLM. {Explanation} and {Question} refer to the generated explanation and the test question, respectively. After obtaining LLM's predictions, we convert the answers through the LLM-predicted Difference Function as shown in Eq. (3).

---

**Algorithm 1** `Raw Differences`

---

**Input:** Two models $f_A, f_B$; condition set $\mathcal{C} = \{c_1, \ldots, c_K\}$; generator $\mathcal{G}$; difference metric $\mathtt{Diff}_{\mathtt{Model}}(f_A, f_B, \cdot)$ from Eq. (2); samples per condition $n$
**Output:** Explanation table $\mathcal{E}$ listing $c_k \mapsto \mathtt{Diff}_{\mathtt{Model}}(f_A, f_B, c_k)$

1: **for** $k = 1$ to $K$ **do**
2:     $\{x_i^c, y_i^c\}_{i=1}^{i=n} \sim \mathcal{G}(c_k)$           $\triangleright$ Generate $n$ samples under condition $c_k$
3:     $\Delta_k \leftarrow \mathtt{Diff}_{\mathtt{Model}}(f_A, f_B, c_k)$           $\triangleright$ e.g., accuracy gap, error rate gap
4: **end for**
5: $\mathcal{E} \leftarrow \{(c_k, \Delta_k)\}_{k=1}^{K}$
6: **return** $\mathcal{E}$

---

**Algorithm 2** `Summarization`

---

**Input:** $\mathcal{E}$ from Alg. 1; Summarization LLM $\mathcal{S}$; prompt template $\pi_{\mathrm{sum}}$
**Output:** Concise natural-language explanation $\hat{\mathcal{E}}$

1: $p \leftarrow \mathrm{FillTemplate}(\pi_{\mathrm{sum}}, \mathcal{E})$
2: $\hat{\mathcal{E}} \leftarrow \mathcal{S}(p)$
3: **return** $\hat{\mathcal{E}}$        $\triangleright$ e.g., "Model A tends to outperform B when the subject is a man . . ."

---

This metric evaluates the completeness, correctness, and sufficiency of an explanation. For instance, when explaining a phenomenon to someone unfamiliar with it, we can assess their understanding by asking a related question. If the explanation is effective, they will be able to provide the correct answer. The choice of the evaluator is critical. In the main paper, GPT-5-mini is employed as a fixed LLM evaluator to provide a consistent assessment of explanation quality across different explanations. Results obtained with alternative LLM evaluators are additionally reported in the Appendix (Experiments section), which further confirms that high-quality explanations remain effective across evaluators.

**`Density`.** We evaluate the counterfactual changes of `Completeness` by applying random dropout to explanation tokens for each question. We use the same above prompt to compute `Completeness` and $25\%$ drop ratio for dropout.

**`Token Length`.** The token count is computed using the *tiktoken* API released by OpenAI. The tokenizer corresponding to the specific model (GPT-5-mini) is employed.

**Attribute Score.** We introduce the attribution score, as shown in Fig. 4. In the CMNIST experiments, the better model under each condition is known. That is, for the question "Which model performs better given $c$?", a ground-truth answer exists (e.g., "Model A"). Using this setup, we can get a loss based on an explanation, a question, and its answer from LLM. To measure token-level contributions, we compute the counterfactual change in loss by removing explanation tokens one at a time (Leave-One-Out): $Loss(\hat{e}) - Loss(e)$. This procedure resembles `Density` (change of completeness), since loss and completeness are inversely related. However, unlike `Density`, it allows fine-grained token-level analysis, albeit at higher computational cost.

## A.2 METHOD

See Alg. 1 for `Raw Differences`, Alg. 2 for `Summarization`, and Alg. 3 for `Optimization`. Below, we describe the prompts used for each method.

**`Summarization`.** Given the results taken from Eq. (2), we provide the below prompt to the summarization LLM.

> You are a machine learning expert. Based on the evaluation results below, explain the strengths and weaknesses of Model A and Model B.
> Requirements:
> - The explanation must be correct and cover all aspects of the given results.
> - Do not simply restate the evaluation results, list pros/cons as is, or include numerical

---

**Algorithm 3** `Optimization`: Iterative Optimization with Feedback and Exploration

---

**Input:** Models $f_A, f_B$; initial conditions $\mathcal{C}_0$; generator $\mathcal{G}$; scorer $\mathcal{Q}$ (Completeness, Density); Feedback & Update LLM $\mathcal{U}$; Exploration LLM $\mathcal{X}$; iterations $T$; samples per condition $n$
**Output:** Optimized explanation $z^\star$
1: $\mathcal{E}_0 \leftarrow \text{RawDifferences}(f_A, f_B, \mathcal{C}_0, \mathcal{G}, n)$            $\triangleright$ Alg. 1
2: $z_0 \leftarrow \text{Summarize}(\mathcal{E}_0, \mathcal{S}, \pi_{\text{sum}})$             $\triangleright$ Alg. 2
3: $s_0 \leftarrow \mathcal{Q}(z_0)$              $\triangleright$ $s$: overall score
4: $z^\star \leftarrow z_0$; $s^\star \leftarrow s_0$; $\mathcal{C} \leftarrow \mathcal{C}_0$
5: **for** $t = 1$ to $T$ **do**
6:   $\mathcal{C}_{\text{new}} \leftarrow \mathcal{X}(z_{t-1}, s_{t-1})$    $\triangleright$ Propose exploratory conditions (e.g., "girl with a doll")
7:   $\mathcal{C} \leftarrow \mathcal{C} \cup \mathcal{C}_{\text{new}}$
8:   $\mathcal{E}_t \leftarrow \text{RawDifferences}(f_A, f_B, \mathcal{C}, \mathcal{G}, n)$
9:   $z_t^{\text{draft}} \leftarrow \mathcal{U}(z_{t-1}, \mathcal{E}_t, s_{t-1})$         $\triangleright$ Edit/Refine the explanation
10:   $s_t \leftarrow \mathcal{Q}(z_t^{\text{draft}})$
11:   **if** $s_t > s^\star$ **then**
12:    $z^\star \leftarrow z_t^{\text{draft}}$; $s^\star \leftarrow s_t$
13:   **end if**
14: **end for**
15: **return** $z^\star$

---

values or result labels.
- Keep it as a single paragraph, concise enough to fit in one or two lines.
- Write in a concise style with short, direct sentences. Avoid unnecessary connectors or long clauses.

Evaluation results:
{List of Model Performance Difference}

Write the explanation in the following format: {"explanation": explanation}.

{List of Model Performance Difference} represents a list of observations, *e.g.*, Model Performance Difference Function.

When we use T2I diffusion models, the possible conditions are open and numerous. In this case, we use LLM to sample the conditions. The below prompt is used for gender experiments.

You are a machine learning expert. Model A and Model B are binary classification models designed to predict gender—either man or woman—from images. Based on the evaluation results below, you need to explain the strengths and weaknesses of Model A and Model B.

To do this, you will identify and list a set of [condition, label] pairs representing different scenarios that could affect model performance. These pairs will be used to further analyze and compare the models' behavior.

Guidelines for the list:
- 'label' should be an integer: '0' for man, and '1' for woman.
- 'condition' should be a brief phrase describing the content of the image, e.g. caption.
- If 'label' is '0', the condition must include a masculine term (e.g., man, boy, etc.).
- If 'label' is '1', the condition must include a feminine term (e.g., woman, girl, etc.).

Please generate exactly {number} pairs in the following list-of-list format: [[condition$_1$, label$_1$], [condition$_2$, label$_2$], ..., [condition$_{\{number\}}$, label$_{\{number\}}$]]. Do not write any additional text outside of the list.

Task description and guidelines for the condition can be adapted for the models' task.

**Optimization.** This method has a feedback/update prompt for Feedback LLM and an exploration prompt for Exploration LLM. Feedback LLM is given the following prompt:

> You are a machine learning expert. Your task is to evaluate an explanation of model results and update it if necessary.
>
> You are given:
> - An explanation of the results
> - {Metrics}
> - New experimental results and updated {Metrics}
>
> Definitions: {Definition of Metrics}
> Requirements for the explanation:
> - The explanation must be correct and cover all aspects of the given results.
> - Do not simply restate the evaluation results, list pros/cons as is, or include numerical values or result labels
> - Keep it as a single paragraph, concise enough to fit in one or two lines.
> - Write in a concise style with short, direct sentences. Avoid unnecessary connectors or long clauses.
>
> Your role:
> 1. Review whether the given explanation sufficiently accounts for the new experimental result.
> 2. Provide feedback on how the explanation can be improved.
> 3. Suggest an updated explanation that integrates both the original points and the new findings, ensuring it is both complete and compact.
>
> Inputs:
> - Explanation: {explanation}
> - {Metrics}: {metric}
> - Experimental result under new condition: {model performance difference}
> - Updated {Metrics}: {update metcis}
>
> Please provide your answer in the following format: "feedback": feedback, "explanation": explanation. Please do not write any additional text outside of the dictionary.

{Metrics} denote the proposed evaluation metrics introduced in the main paper. In practice, we may provide either a single metric or the full metrics. The definitions are given to the LLM to ensure that it can interpret the intended meaning of each metric. Given both the previous metric values and the updated ones under the new result of model performance differences, the LLM is required to reason about how the explanation should be improved with respect to these metrics. Exploration LLM is given the following prompt:

> You are a machine learning expert. You wrote the explanations that describe the strengths and weaknesses of two models, Model A and Model B. Below are the scores of your explanations based on {Metrics}.
>
> {Definition of Metrics}
>
> History:
> {Explanation 1}: {explanation 1}
> {Metricsc 1}: {metrics 1}
>
> {Explanation 2}: {explanation 2}
> {Metricsc 2}: {metrics 2}

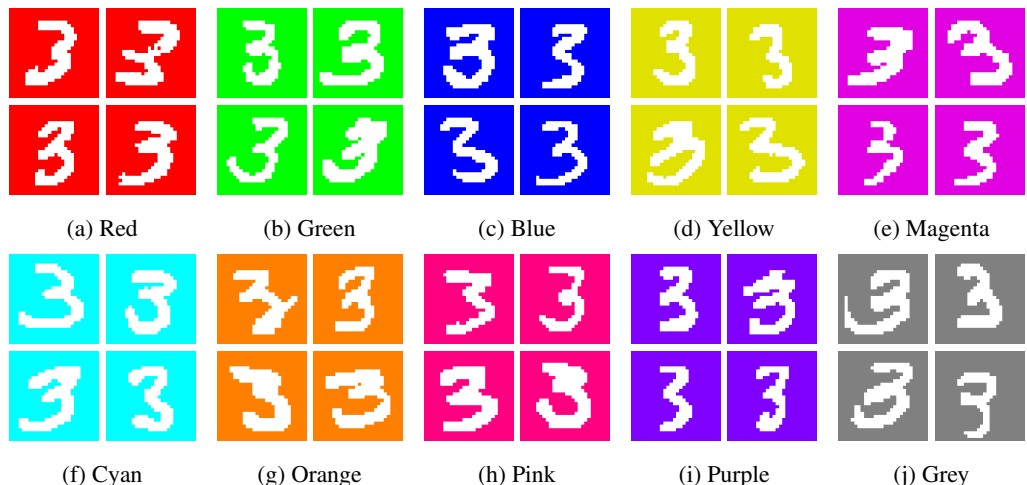

Figure A: CMNIST color examples.

{Explanation 3}: {explanation 3}
{Metricsc 3}: {metrics 3}

To improve the scores of these explanations, you need to gather additional information by exploring further conditions:
{Description of Conditions}

The chosen values should be those you consider the most important to explore further, and they must fall within the given ranges. Additionally, you must decide whether the strategy is "exploration" (searching new conditions broadly) or "exploitation" (focusing on promising conditions).

Please output the conditions in the following list-of-dictionaries format: [{strategy: exploration or exploitation, {condition: condition}, ...]. Please provide {exploration size} different conditions, and do not write any additional text outside of the list.

This prompt is designed to guide an LLM in simulating the process of improving explanations for comparing two models (Model A and Model B). The LLM is first provided with prior explanations and their corresponding scores under the proposed {Metrics}. To ensure proper interpretation, the definitions of the metrics ({Definition of Metrics}) are explicitly given. With this context, the LLM is tasked with enhancing explanation quality by identifying additional conditions that could reveal further insights. The prompt supplies a description of available conditions ({Description of Conditions}). The LLM must select the conditions it considers most important to explore further and assign a strategy of either exploration (broadly searching new conditions) or exploitation (focusing on promising conditions).

## A.3 DATA

We need two pre-trained models to be compared. We prepare the training data, the models, and the evaluation set as described in the main paper.

**CMNIST.** The MNIST dataset (LeCun et al., 2010) is a handwritten digits dataset widely used in machine learning research. It is composed of 28×28 handwritten digits (0–9). CMNIST (Arjovsky et al., 2019; Bahng et al., 2020) introduces color bias by associating each digit with a specific background color (e.g., digit 0 with red). For two models to be compared, each model is trained on different color biases explicitly. The model has the same CNN architecture: four $7 \times 7$ convolutional layers, batch normalization (Ioffe & Szegedy, 2015), and ReLU. The colors are chosen in

red, green, blue, yellow, magenta, cyan, orange, pink, purple, and grey. See Fig. A for the color examples.

The test questions cover all combinations of a digit, a color, and a rotation. Rotation does not affect the performance compared to the color and acts as a confusing factor. We have 10 digits, 10 colors, and 13 degrees, so the total number of test questions is 1,300.

**CLEVR.** We employ the CLEVR dataset (Johnson et al., 2017), which consists of synthetic images generated using Blender. Each image is characterized by four attributes: shape, material, color, and size. Specifically, the dataset includes three shapes (cube, cylinder, sphere), two materials (metal, rubber), eight colors (gray, red, blue, green, brown, purple, cyan, yellow), and three sizes (small, medium, large). To facilitate comparison, we train two models with explicitly imposed material biases. The CNN architecture used is identical to that employed for CMNIST. The test set comprises a total of 144 questions.

**Gender Dataset.** For gender prediction, we use SigLIP, a zero-shot image classification model guided by text prompts. Model A has the prompts: *"a photo of a man with black hair"* and *"a photo of a woman with blond hair"*. Model B has the prompts: textit"a photo of a man with blond hair" and *"a photo of a woman with black hair"*. Since hair color and gender are correlated, we hypothesize that Model A is more biased toward gender cues than Model B.

To measure `Sufficiency`, we need to test questions and corresponding images. As mentioned in the main paper, we generate the synthetic dataset using Stable Diffusion 3.5 (Esser et al., 2024). Males and females have 50 captions for conditions, respectively. Thus, the total number of text questions is 100. Texts used to generate the synthetic dataset are listed below.

> A lone {male or female} traveler walking across a desert at sunset
> A smiling {male or female} chef cooking in a cozy kitchen
> A {male or female} ballerino mid-twirl on an empty stage
> A solitary {male or female} knight standing in a misty forest
> A young {male or female} scientist working late in a lab
> A {male or female} fisherman casting a line into a calm lake at dawn
> A {male or female} business professional presenting in a modern office
> A {male or female} painter creating a colorful mural on a blank wall
> A {male or female} street musician playing violin under a lamppost
> A {male or female} yoga instructor meditating on a mountain peak
> A medieval {male or female} archer aiming at a target in a clearing
> A futuristic {male or female} soldier in armor on a dystopian street
> A {male or female} librarian reading quietly in an ancient library
> A {cowboy or female cowboy} riding a horse across an open plain
> A {male or female} deep-sea diver swimming near coral reefs
> A teenage {boy or girl} skateboarding down an empty road
> A {male or female} writer typing intensely in a cluttered study
> A {male or female} monk praying inside an ancient temple
> A {male or female} singer performing passionately on a lit stage
> A {male or female} firefighter standing heroically amidst smoke
> A {male or female} fashion model posing on a minimalist set
> A {male or female} gardener tending flowers in a sunny backyard
> A {male or female} pilot in uniform walking across a runway
> A {male or female} astronaut with his helmet off floating inside a space station
> A {male or female} swordsman practicing under cherry blossoms
> A {male or female} mountain climber reaching the summit alone
> A {male or female} mechanic fixing a car in a dimly lit garage
> A {male or female} police officer directing traffic at a busy crossing
> A {male or female} student studying alone in a library at night
> A {male or female} surfer riding a massive wave at sunset
> A {male or female} samurai standing in a bamboo forest
> A {male or female} poet reciting verses by a riverside
> A {male or female} detective inspecting a crime scene at night

*"A {male or female} baker decorating a cake in a colorful bakery."*     *"A {male ballerino or ballerina} mid twirl on an empty stage."*

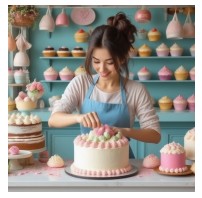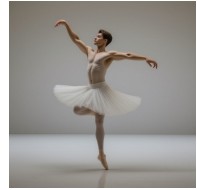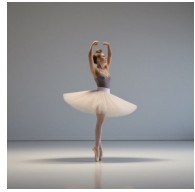

*"A {male or female} firefighter standing heroically amidst smoke."*     *"A {male or female} samurai standing in a bamboo forest."*

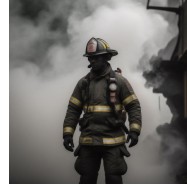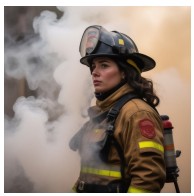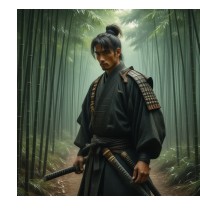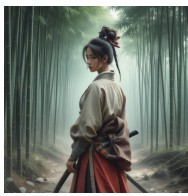

Figure B: **Samples of gender synthetic dataset samples.**

A {male or female} farmer harvesting crops under a bright sky
A {male or female} violinist practicing in a grand concert hall
A {male or female} boxer training alone in a gym
A {male or female} baker decorating a cake in a colorful bakery
A {male or female} priest giving a sermon in an empty cathedral
A {male or female} doctor examining an X-ray in a quiet office
A {male or female} sailor steering a boat through foggy waters
A {male or female} carpenter building furniture in a sunlit workshop
A {male or female} sorcerer casting spells in a dark forest
A {male or female} fashion designer sketching new outfits
A {male or female} wizard studying ancient scrolls in a stone tower
A {male or female} robot engineer assembling a humanoid android
A {male or female} jazz musician playing saxophone in a smoky bar
A {male or female} skier descending a snowy mountain slope
A {male or female} dancer practicing moves in a mirrored studio
A {male or female} photographer setting up a tripod on a beach
A {male or female} biologist examining plants in a dense rainforest

Under the above conditions, we generate image samples for Eq. (2); we also remove ambiguous images and re-generate images if necessary. The number of image samples for each condition is 8, so the number of total images is 800. Figure B shows the generated image samples.

**CelebA.** CelebA (Liu et al., 2015) is a large-scale facial attribute dataset. The dataset is available for non-commercial research purposes only. We use it to evaluate the impact of initialization and to demonstrate the application of textual explanations.

### A.4 EXPERIMENT

**Experiments compute resources.** We conduct our experiments using A6000, A100 (40GB), and V100 GPUs on a Slurm-based infrastructure. Our method runs on a single GPU by leveraging efficient LLM inference techniques, such as 8-bit or 4-bit quantization.

**Hyperparameters.** We fix the random seed for reproducibility. For explanation generation, we use LLMs with a temperature of 1.0 to allow non-deterministic outputs. All other hyperparameters follow the default settings of the Hugging Face API. During evaluation, we use LLMs in a deterministic setting. Additionally, we set the number of explanations to be kept during explanation generation to three.

| | |
|---|---|
| Human 1 | Model A is better than model B when the digits of 0, 1, 2, 3, and 4 are colored in red regardless of the rotation angle, but not in other colors. Model B is better than model A when the digits of 5, 6, 7, 8, and 9 are colored in blue regardless of the rotation angle, but not in other colors. |
| Human 2 | Model A is worse than model B when the digits of 0, 1, 2, 3, and 4 are not colored in red regardless of the rotation angle. Model B is worse than model A when the digits of 5, 6, 7, 8, and 9 are not colored in blue regardless of the rotation angle. |
| Human 3 | Model B is better than model A for the digits 0, 1, 2, 3, and 4 in any color other than red regardless of the rotation angle. Model A is better than model B for the digits 5, 6, 7, 8, and 9 in any color other than blue regardless of the rotation angle. |
| Summarization (Llama 3.1) | Model A performs well when the digit is 9, color is magenta or grey, and angle is within a certain range. However, Model A struggles with digits 0, 4, and 2, especially when the color is yellow, green, or cyan, and the angle is outside of a specific range. Model B performs better with digits 0, 4, and 2 in various color and angle combinations, but its accuracy drops when the digit is 9, color is magenta or grey, and angle is within a specific range. Both models have strengths and weaknesses, and their performance varies depending on the input conditions. |
| Optimization (Llama 3.1) | Model A and Model B show varied performance under different conditions. While A underperforms with digit 1 and certain angles, it outperforms with digits 5, 6, 7, and 8. Model B shows relative stability with certain digits and conditions but underperforms with others. |
| Summarization (Phi 4) | Model A excels in scenarios involving digits 9, 5, and 7 with specific color and angle combinations, showing strong performance in these contexts. Conversely, Model B performs better with digits 0, 4, and 3, particularly when certain colors and angles are present. Model A struggles with digits 1 and 4 across various conditions, while Model B shows weaknesses with digits 9 and 8 in specific settings. Both models have instances of equal performance, indicating potential areas of overlap in their capabilities. |
| Optimization (Phi 4) | Model A excels with digits 9 and 7 in specific conditions, while Model B outperforms with digits 4, 0, 1, 2, 3, and 5 in various colors and angles. Model A also excels with digit 8 in orange at -30 degrees and digit 6 in orange at 5 degrees. |

Figure C: **Generated Explanation on CMNIST.** We offer three explanations using the knowledge of how to train Model A/B, which are the upper bound. Human 2 is used in the main paper.

| | |
|---|---|
| Summarization (Llama 3.1) | Model A performs well when classifying cubes with various colors and materials, especially when the size is large. Model A also excels with spheres of certain colors and materials, particularly when the size is large. However, Model A struggles with cylinders, especially when the material is metal and the size is large. Model B, on the other hand, struggles with spheres of certain colors and materials, particularly when the size is small, and performs poorly with cylinders of various colors and materials when the size is large. Model B has some strengths with cylinders of specific colors and materials when the size is medium, but these are not consistent across all evaluations. |
| Optimization (Llama 3.1) | Model A excels at classifying spheres, especially when they are large and made of rubber, but struggles with cylinders, particularly those with varying colors and materials. Model B performs well with cylinders, especially when they are medium-sized and made of metal, but falters with spheres, especially when they are small and made of rubber. Overall, both models have strengths and weaknesses, suggesting that they are complementary and could be used in conjunction to improve overall accuracy. |
| Summarization (Phi 4) | Model A excels with spheres, particularly in specific color-material-size combinations, and performs well with large gray rubber cubes, while Model B is superior with blue rubber cubes and medium green rubber cubes. Model A struggles with brown rubber cylinders and large brown rubber cylinders, whereas Model B shows consistent performance with metal cylinders and rubber cubes of various colors and sizes. |
| Optimization (Phi 4) | Model A excels with small, cyan spheres and medium, green or gray cubes, while Model B performs better with large, blue, metallic cylinders or medium-sized, brown or cyan cylinders. Model A struggles with large, yellow cylinders or medium-sized, brown ones, whereas Model B is consistent with cylinders. New results show Model A outperforms Model B with large, red, rubber spheres, while Model B excels with small, purple, metal cubes and medium, gray, rubber cylinders. Model A also outperforms Model B with large, brown, rubber cubes, but both models perform equally with medium, cyan, metal cylinders and large, red, metal cubes. |

Figure D: **Generated Explanation on CLEVR.**

## B  ADDITIONAL EXPERIMENT RESULTS

### B.1  GENERATED EXPLANATIONS

We provide qualitative results of the generated explanations for the CMNIST (Fig. C), CLEVR (Fig. D), and Gender classification experiments (Fig. E).

### B.2  OTHER LLM EVALUATOR

Table A reports completeness scores when alternative LLM evaluators (Llama 3.1 and Phi 4) are used in place of GPT-5. While the overall trends remain consistent—optimization-based methods generally outperform summarization—the absolute scores differ across evaluators. For instance, Llama 3.1 and Phi 4 sometimes assign higher `Completeness` values to `Summarization` compared to GPT-5. These variations suggest that evaluator choice can influence the absolute scale of scores. For consistency and reliability, we therefore use GPT-5 as the primary evaluator in the main experiments.

| | |
|---|---|
| Summarization (Llama 3.1) | Model A tends to perform better than Model B when the subject is a man, especially in scenarios where men are dressed formally or have distinct facial features, such as a beard or mustache. However, Model A struggles to accurately classify pregnant women and women with babies, indicating potential biases in its training data. Model B appears to be more accurate in these cases, but its overall performance is weaker compared to Model A, particularly when the subject is a man in a specific context or with distinct facial features. |
| Optimization (Llama 3.1) | Model A outperforms Model B when the image condition involves men in professional or stereotypically masculine contexts, such as a man in a suit, man holding a briefcase, or man with a beard. In contrast, Model B performs better in scenarios involving women or children, such as a girl playing with dolls, woman with a purse, or woman with a baby. Model A and Model B perform equally well in conditions with neutral or ambiguous gender cues, like a boy riding a bike, young girl smiling, or boy playing soccer. |

Figure E: **Generated Explanation on Gender.**

Table A: **Other Evaluator Results on CMNIST.**

| LLM | Method | Completeness | | |
|---|---|---|---|---|
| | | GPT 5 | Llama 3.1 | Phi 4 |
| - | Human 1 | 0.00 | -0.16 | 0.00 |
| - | Human 2 | 0.90 | 0.06 | 0.69 |
| - | Human 3 | 0.90 | 0.88 | |
| - | Raw Differences | 0.33 | 0.50 | - |
| Llama 3.1 8B | Summarization | 0.55 | 0.50 | 0.42 |
| | Optimization | 0.66 | 0.66 | 0.46 |
| Phi 4 14B | Summarization | 0.58 | 0.70 | 0.80 |
| | Optimization | 0.67 | 0.62 | 0.67 |

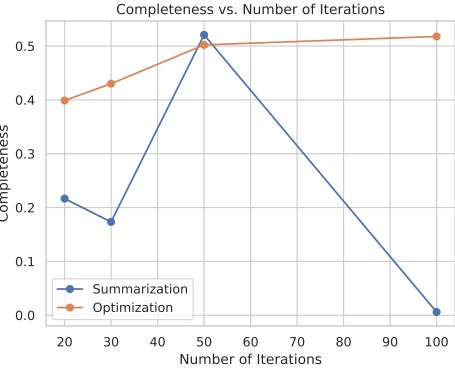

Figure F: **Completeness With Varying Numbers of Iterations on CLEVR**. The optimization-based method exhibits stable improvements as iterations increase, whereas the summarization-based method fluctuates and even degrades after peaking around 50 iterations.

| | |
|---|---|
| 20 Iteration | Model A excels at classifying spheres, especially when they are large and made of rubber, but struggles with cylinders, particularly those with varying colors and materials. Model B performs well with cylinders, especially when they are medium-sized and made of metal, but falters with spheres, especially when they are small and made of rubber. Overall, both models have strengths and weaknesses, suggesting that they are complementary and could be used in conjunction to improve overall accuracy. |
| 30 Iteration | Model A excels with rubber shapes, especially spheres and small cubes, but struggles with metal shapes. Model B performs better with metal shapes, particularly cylinders, but underperforms with rubber shapes. New results show Model A's strong performance with rubber spheres and cubes, but Model B's edge with metal cylinders and rubber cubes. |
| 50 Iteration | Model A excels in cube classification, especially with rubber and large sizes, and shows an advantage in some sphere scenarios. Model B performs well in sphere classification, particularly with metal and specific color combinations. However, both models have varying performance under different conditions, with some scenarios showing no difference or even a disadvantage for Model A, as seen in the new experimental results where accuracy differences range from 0.0% to 100.0%. |
| 100 Iteration | Model A excels with uncommon combinations and improves with certain color-material pairs, but struggles with small shapes and specific color-material combinations. Model B performs well with typical metal shapes and standard sizes, but struggles with rubber shapes and non-standard sizes. The accuracy difference between models varies with specific attributes, with some combinations showing significant gaps in performance. |

Figure G: **Generated Explanation from CLEVR with More Iterations**

### B.3 MORE ITERATION

As shown in Fig. F, `Completeness` under the optimization-based approach steadily improves as the number of iterations increases, indicating stable refinement. In contrast, summarization fluctuates considerably and even declines after 50 iterations, suggesting limited robustness to iteration scaling. Figure G shows the generated explanations.

