# OpenReview forum: "Towards Automatic Discovery and Explanation of Differences Between Vision Models"
_ICLR.cc/2026/Conference — ICLR 2026 Conference Withdrawn Submission_

### Official Review · Reviewer_azL8 · 2025-10-19

**Soundness:** 3
**Presentation:** 2
**Contribution:** 2
**Rating:** 2
**Confidence:** 4

**Summary:**

This paper proposes to explain the differences between two vision models using natural language, using an LLM and a conditional data generator. They propose 3 evaluation metrics for the explanation: Completelness, Density and Token Length. They also derive three explanations: Raw differences enumerates a list of all conditions, but is too much information, while Summarization is a brief summary of all conditions but may omit details. Optimization is based on steering an LLM with the objectives of having both good Raw Differences and Summarization. Experiments on CMNIST, CLEVR, and CelebA show that the Optimization method produces the most concise and informative explanations, effectively uncovering true model differences

**Strengths:**

- Explanations between the differences of two models via natural language is an interesting and unstudied direction.
- The motivation is well defined and is interesting.

**Weaknesses:**

- [W1] The paper does not investigate realistic scenarios. Most of the cases are binary cases (e.g., gender), based on toy, unrealistic datasets such as CMNIST and CLEVR, and even with those, very controlled experiments where authors train their own models on their own bias conditions. I am not really sure how this paper would have an impact in the field. On the other hand, GLOVE [R1] is a similar work in essence, that also uses LLMs as implicit optimizers to generate input prompts for vision-language models. Here, the authors "explanations" can be viewed as input prompts. Furthermore, GLOVE shows how useful the method is on a variety of real-word tasks and models, such a CLIP, and autoregressive VLMs, across a huge variety of tasks including multi-label zero-shot classification, generalization and VLM safety.

- [W2] I believe that simply designing datasets, models, and conditions, and then evaluating them on self-proposed metrics without demonstrating their broader applicability, offers limited value to the community. While such metrics can indeed serve as useful reward signals for guiding LLMs (as in GLOVE), reporting them in isolation—without showing their practical relevance or impact—does not meaningfully advance the field.

- [W3] The explanations are highly built on synthetic data, which might differ from the real data distribution that the model is trained on. How do the authors fix this issue? This leads to completely misleading explanations if that is the case.

[R1] GLOV: Guided Large Language Models as Implicit Optimizers for Vision Language Models, TMLR 2025

**Questions:**

While the idea is nice and interesting, W1 and W2 are major and sadly enough, my decision will be negative. I encourage the authors to continue on the same direction, but show practicality and usefulness of their method in real-word settings and tasks.

Minor:

- The figures are misleading. For eg. L240 and Fig 3. "man with a beard", "woman with a stroller"...etc are not attributes, you explicitly specify the gender. L246 "the explanation becomes lengthy" is not reflected in the Figure.
- I recommend citing related work on explaining the difference between two models via concepts [R2].


[R2] REPRESENTATIONAL SIMILARITY VIA INTERPRETABLE VISUAL CONCEPTS, ICLR 2025

**Details Of Ethics Concerns:**

No issues

---

### Official Review · Reviewer_TLLd · 2025-10-29

**Soundness:** 2
**Presentation:** 2
**Contribution:** 2
**Rating:** 2
**Confidence:** 4

**Summary:**

This paper proposes a method to automatically identify and explain performance differences between machine learning models beyond benchmark scores. They introduce a framework that generates natural language explanations describing the differences between two models’ performances.

To evaluate explanation quality, they define three metrics: Completeness (measuring correctness and overall informativeness), Density (capturing token-level informativeness), and Token Length (assessing verbosity). Based on these metrics, three explanation generation methods are proposed: Raw Differences, which enumerates all performance differences; Summarization, which condenses them into concise summaries; and Optimization, which balances informativeness and conciseness.

Experiments on the CMNIST, CLEVR, and CelebA datasets demonstrate that the Optimization method effectively reveals model differences and biases through natural language.

**Strengths:**

The paper introduces three evaluation metrics—Completeness, Density, and Token Length—along with three complementary methods to analyze performance differences between models.

The results demonstrate that the Summarization and Optimization methods generate more concise and informative explanations, effectively capturing high-level insights.

Moreover, the authors validate the robustness and generality of their approach using different large language models (LLMs), further confirming the effectiveness of their proposed framework.

**Weaknesses:**

Although the proposed framework aims to reduce the extensive human effort, time, and resources required to identify model strengths and weaknesses, the paper does not provide quantitative comparisons with existing approaches to substantiate this claim.

In addition, the evaluation is conducted on only 100 conditions, which is insufficient to yield strong quantitative evidence or confidently demonstrate the framework’s effectiveness.

While experiments are performed on three datasets—CMNIST, CLEVR, and CelebA—these datasets are relatively limited in scope, as they represent simple numerical, synthetic, and facial data, respectively. Evaluating the method on larger and more diverse datasets, such as ImageNet, could provide a stronger benchmark and better validate generalization.

Furthermore, the paper lacks clarity regarding the models used for training on these datasets, including whether they share the same architecture or differ, which makes it difficult to interpret the reported comparisons.

**Questions:**

It remains unclear why the study did not include direct comparisons between existing visual models to further validate the proposed framework. This would be particularly interesting, as numerous prior works have explored differences between vision models.

1. Rulin Shao, Zhouxing Shi, Jinfeng Yi, Pin-Yu Chen, and Cho-Jui Hsieh. On the adversarial robustness of visual transformers. arXiv preprint arXiv:2103.15670, 2021.

2. Yutong Bai, Jieru Mei, Alan L Yuille, and Cihang Xie. Are transformers more robust than cnns? Advances in Neural Information Processing Systems, 34, 2021.

3. Mingqi Jiang, Saeed Khorram, and Li Fuxin. Comparing the decision-making mechanisms by transformers and cnns via explanation methods. In IEEE Conf. Comput. Vis. PatternRecog. (CVPR), pages 9546–9555, 2024.

---

### Official Review · Reviewer_dtVL · 2025-10-31

**Soundness:** 1
**Presentation:** 2
**Contribution:** 1
**Rating:** 2
**Confidence:** 4

**Summary:**

This paper proposes using large language models to explain the differences between two models through three steps: Raw Differences, Summarization, and Optimization. The approach leverages LLMs to generate comparative explanations. In addition, the authors introduce three metrics to evaluate the quality of the explanations: Completeness, Density, and Readability. Experimental results show that the proposed method outperforms baseline approaches on multiple datasets.

**Strengths:**

1. The method is simple: by leveraging the capabilities of large language models, it generates comparative explanations without requiring complex training processes.

**Weaknesses:**

1. Based on Definitions 1.1 and 1.2, the task can be reduced to explaining a binary classifier and using the explanation to predict the model’s output. One could simply define a difference model ( f_{\text{diff}} = f_1 - f_2 ) and apply existing explanation methods to ( f_{\text{diff}} ). Since the paper does not compare against this natural baseline in the related work or experimental evaluation, the effectiveness of the proposed method remains unconvincing.
2. The paper primarily uses Raw Differences as a baseline but does not clarify how the Conditions \(c\) for generating Raw Differences is chosen. The quality of Raw Differences is highly sensitive to the choice of \(c\)s. Intuitively, selecting an appropriate condition could significantly improve the results. In Section 4, the authors draw an analogy between conditions and concepts. However, existing methods for concept selection[1] already provide effective ways to identify concepts that strongly influence model predictions, which could also improve Raw Differences. The lack of comparison with such established methods further weakens the paper’s claims.
3. The authors introduce three new metrics for evaluating explanation quality. However, evaluation of explanations is already a well-studied problem[2-4]. Without comparison to existing evaluation metrics, presenting these three as a main contribution is inadequate.

[1] Concept-based Explainable Artificial Intelligence: A Survey

[2] A comprehensive study on fidelity metrics for XAI

[3] XForecast: Evaluating Natural Language Explanations for Time Series Forecasting

[4] On the (in) fidelity and sensitivity of explanations

**Questions:**

1. How are the Conditions or concepts (c) chosen when generating Raw Differences? Would different choices of ( c ) significantly affect the quality of the results?
2. How does the proposed method compare to the alternative approach of transforming the problem into explaining a single difference model ( f_{\text{diff}} ) using existing explanation methods?

---

### Official Review · Reviewer_ZbTk · 2025-10-31

**Soundness:** 2
**Presentation:** 3
**Contribution:** 2
**Rating:** 4
**Confidence:** 4

**Summary:**

This paper introduces a framework for automatically generating natural language explanations that describe performance differences between two vision models. It tackles a very important explainability problem in vision models. while there could be gaps in terms of performance and reliability.

**Strengths:**

- The paper introduced well-defined and complementary metrics, such as completeness, density, and token length, that are reasonable for evaluating explanations.
- It performed ablation studies to understand each design choice’s impact on metrics.
- A practical use case on CelebA was shown.

**Weaknesses:**

- While the paper used a generator to produce images for model evaluation, it has not been validated whether the generated images are diverse and realistic.
- The conditions proposed by LLMs may be prone to errors and may not be diverse.
- There is a significant gap between human and LLM-generated explanations in Table 1. The practical implications of the proposed method should be more clearly discussed.

**Questions:**

- How sensitive are the results to the quality and diversity of the images/conditions generated by the underlying data generators and the LLM for exploration? Have any analyses been conducted regarding potential biases or limitations in the synthetic data or LLM-proposed conditions?
- What was the computational cost of the method?

---

### Note · Authors · 2025-11-18

I have read and agree with the venue's withdrawal policy on behalf of myself and my co-authors.